# Identification of a New and Effective Marker Combination for a Standardized and Automated Bin-Based Basophil Activation Test (BAT) Analysis

**DOI:** 10.3390/diagnostics14171959

**Published:** 2024-09-04

**Authors:** Johannes Groffmann, Ines Hoppe, Wail Abbas Nasser Ahmed, Yen Hoang, Stefanie Gryzik, Andreas Radbruch, Margitta Worm, Kirsten Beyer, Ria Baumgrass

**Affiliations:** 1German Rheumatology Research Center (DRFZ), A Leibniz Institute, 10117 Berlin, Germany; 2Department of Rheumatology and Clinical Immunology, Charité—Universitätsmedizin Berlin, 10117 Berlin, Germany; 3Division of Allergy and Immunology, Department of Dermatology, Venerology and Allergy, Charité—Universitätsmedizin Berlin, 10117 Berlin, Germany; 4Department of Pediatric Respiratory Medicine, Immunology and Critical Care Medicine, Charité—Universitätsmedizin Berlin, 13353 Berlin, Germany; 5Institute of Biochemistry and Biology, Faculty of Science, University of Potsdam, 14476 Potsdam, Germany

**Keywords:** basophil activation test, multi-parametric analysis, re-analysis, combinatorial protein expression, high-dimensional cytometry data, mass cytometry data

## Abstract

(1) Background: The basophil activation test (BAT) is a functional whole blood-based ex vivo assay to quantify basophil activation after allergen exposure by flow cytometry. One of the most important prerequisites for the use of the BAT in the routine clinical diagnosis of allergies is a reliable, standardized and reproducible data analysis workflow. (2) Methods: We re-analyzed a public mass cytometry dataset from peanut (PN) allergic patients (*n* = 6) and healthy controls (*n* = 3) with our binning approach “**p**attern **r**ecognition of **i**mmune cells” (PRI). Our approach enabled a comprehensive analysis of the dataset, evaluating 30 markers to achieve optimal basophil identification and activation through multi-parametric analysis and visualization. (3) Results: We found FcεRIα/CD32 (FcγRII) as a new marker couple to identify basophils and kept CD63 as an activation marker to establish a modified BAT in combination with our PRI analysis approach. Based on this, we developed an algorithm for automated raw data processing, which enables direct data analysis and the intuitive visualization of the test results including controls and allergen stimulations. Furthermore, we discovered that the expression pattern of CD32 correlated with FcεRIα, anticorrelated with CD63 and was detectable in both the re-analyzed public dataset and our own flow cytometric results. (4) Conclusions: Our improved BAT, combined with our PRI procedure (bin-BAT), provides a reliable test with a fully reproducible analysis. The advanced bin-BAT enabled the development of an automated workflow with an intuitive visualization to discriminate allergic patients from non-allergic individuals.

## 1. Introduction

According to the European Academy of Allergy and Clinical Immunology (EACCI), allergy is the most common chronic disease in Europe affecting more than 150 million Europeans [1]. This highlights the need for better public health measures, research and improved diagnostic capabilities. The basophil activation test (BAT) is a functional ex vivo assay that uses whole blood to measure basophil activation in response to allergen exposure, utilizing flow cytometry. It can complement the skin prick test (SPT) and the specific IgE (sIgE) test by providing a more comprehensive, sensitive and specific assessment of allergic reactions, especially in complex cases. BAT can also recognize some allergic reactions in which no IgE antibodies are involved [2]. Often, BAT helps to avoid provocation tests, such as the oral food challenge (OFC), by identifying allergens without exposing patients to potentially severe reactions [3,4,5].

However, there are several unresolved BAT issues related to the flow cytometric measurement and analysis of multi-dimensional single-cell data. This includes the lack of general availability and acceptance of standardized protocols, standardized controls and reference ranges despite joint efforts and recommendations [6]. Others are technical complexity due to specialized equipment and critical technical expertise and data analysis. Analyzing flow cytometry data is complex and requires sophisticated software and expertise. A major problem in data analysis is reproducibility, as the variability of basophil gating strategies and the interpretation of activation markers is user-dependent and can lead to inconsistent results.

Recently, we developed a new bin-based approach called “pattern recognition of immune cells” (PRI) for the reproducible analysis and visualization of cytometric data [7,8,9]. This PRI approach utilizes binning to facilitate feature engineering through the combination of three or more protein markers. Additionally, this method generates bin plots, offering various statistical data analyses and a semi-continuous visualization of marker intensities that can reveal characteristic patterns of immune cell populations. Here, we used this approach to open new insights from a public mass cytometry dataset of 30 markers from peanut (PN) allergic patients [10,11]. The newly identified marker combination consisted of the two Fc-receptors FcεRIα and FcγRII and the IL-3 receptor CD123 for basophil identification and the activation marker CD63 to monitor basophil degranulation. The integration of these markers into our bin-based analysis approach allowed us to identify activated basophils in a fully reproducible manner using flow cytometry. This new method has been further developed into an automated bin-BAT workflow and can now be used to diagnose patients with allergies.

## 2. Materials and Methods

### 2.1. Study Approval, Data Sets and Study Populations

The study was approved by the Charité’s Ethics Committee in Berlin, Germany (EA2/203/21 and EA2/304/21). All participants in the study provided written informed consent.

Re-analysis, shown in the first two figures, was performed on a public mass cytometry dataset from the database FlowRepository [10,11].

Our own data for this study, shown in the last two figures, was collected from participants recruited from 2021 to 2024 in Berlin, Germany. Adult patients with allergies, included in the dataset in the third subfigure of figure three, were allergic to food and inhalation allergens. Allergic status was defined by recent positive allergy test results, such as the oral food challenge (OFC; in 38.8% of all patients with allergies), basophil activation test (BAT; 89.8%), skin prick test (SPT; 79.6%) or specific Immunoglobulin E (sIgE; 51.0%). All patients with allergies reported allergic symptoms. No allergy medications were taken before blood withdrawal. Non-allergic controls exhibited negative allergy test results and no reported symptoms. Exclusion criteria for this dataset involved unclear allergic status, like conflicting allergy test results and symptoms, as well as IgE non-responders, whose basophils fail to activate in response to IgE-mediated triggers in the BATs. The dataset for the last subfigure also included child patients who were either allergic or tolerant without clinical relevance.

### 2.2. Flow Cytometric Instruments and Antibodies

All measurements were conducted on a BD LSR Fortessa (BD Biosciences, San Jose, CA, USA) or MACSQuant10 and MACSQuant16 (Miltenyi Biotec, Bergisch Gladbach, Germany). Basophil activation was measured by staining whole blood with antihuman CD63-VioBlue (clone: H5C6, Miltenyi Biotec) before fixation. The following antihuman antibodies were used after fixation: CD123-FITC (clone: AC145, Miltenyi Biotec), FcεRIα-PE-Vio 770 (clone: CRA1, Miltenyi Biotec), HLA-DR-PerCP-Cy5.5 (clone: G46-6, BD Biosciences), CD32-Alexa Fluor 647 (clone: FUN-2, BioLegend, San Diego, CA, USA).

### 2.3. Basophil Activation Test

Fresh heparinized whole blood was mixed with RPMI 1640 Medium, GlutaMAX™ (Thermo Fisher Scientific, Waltham, MA, USA) with a ratio of 1:1 or 5:2, preheated to 37 °C, containing 250 µg/mL CD63 antibody and the allergen extract or control. Positive control was 0.25 µg/mL αImmunoglobulin E (αIgE) (Bethyl Laboratories, Montgomery, AL, USA), negative control was phosphate buffered saline (PBS) and allergen stimulation was 1 µg/mL birch pollen extract (DST GmbH, Schwerin, Germany). No IL-3 for costimulation was used except for the first two subfigures of figure three (Miltenyi Biotec, 2 ng/mL). The stimulation mix was incubated for 15 min in a water bath at 37 °C. The stimulation was stopped by adding BD Phosflow Lyse/Fix Buffer (BD Biosciences), followed by two washing steps with phosphate-buffered saline containing 0.2% BSA (PBS/BSA). Cells were stained using the basophil-specific antibodies FcεRIα and CD32, and during method establishment also CD123 and HLA-DR. After further washing, the samples were analyzed.

### 2.4. Manual Flow Cytometry Data Analysis

Data analysis was performed using FlowJo™ v10.10 Software (BD Life Sciences, Ashland, OR, USA) for pregating on single cells using different scatter parameters. Eosinophils were outgated based on their scatter properties to exclude autofluorescent cells (first three figures). Further analysis was performed using the PRI analysis method, which was previously described [7].

### 2.5. Development of an Automated BAT Analysis Tool Using the PRI Approach

Our novel workflow to analyze and visualize BAT-raw data uses the multi-parametric binning of our PRI algorithm (for detailed explanations of PRI [8]). The automatic analysis workflow is based on Python [12], including pregating and plotting using R (pregating functions are based on Bioconductor tools [13,14]), and plotting is performed using R-core [15]. The accuracy of the auto-BAT was validated by comparing its results with the respective manually analyzed results.

The workflow involves the following fully automated steps:Pregating:

Single cell gating to exclude doublets, thrombocytes and cell debris from all files.

Basophil identification:

Threshold setting for side scatter (SSC) and FcεRIα (x- and y-axis), using CD32 as an auxiliary marker to isolate basophils in the upper left quadrant and CD63 (z-axis) to define 1% as positive in each patient’s unstimulated sample. The same set of thresholds are applied to all other samples

User-friendly result output:

Analysis results are stored in a database from which bin plot compilations and Excel files can be retrieved for each experiment. Furthermore, the entire data set can be analyzed and queried in its entirety. A system of quality control messages is provided with each bin-plot enabling a reliable analysis workflow.

A flow chart of the automatic workflow and a detailed description of the process are provided in Appendix A.

### 2.6. Quantification and Statistical Analysis

Statistical analysis was conducted using GraphPad Prism v10.2.2 (Boston, MA, USA). Data for the third subfigure of figure three was analyzed using CD32 thresholds based on red percentages for basophils in the upper left quadrant set to 70% for each patient.

The median was calculated from CD32 thresholds of non-allergic patients or patients with allergies within one experiment. A Wilcoxon matched-pairs signed rank test was applied to compare the CD32 threshold median values between non-allergic patients and patients with allergy in each experiment because of the pairwise data structure, not normally distributed data and sufficient sample size for robust results interpretation.

A value of *p* < 0.05 was considered statistically significant.

## 3. Results

### 3.1. Re-Analysis of Mass Cytometry Data Sets Discovered a Useful New Marker Combination for BAT

#### 3.1.1. PRI-Based Identification of the Best Basophile Identification Marker Combination

Our PRI approach enabled us to select the optimal marker combination, FcεRIα^high^ and CD123^+^, from a dataset of 30 markers to achieve the excellent separation of basophils from all other blood cells (Figure 1). The bin-based PRI grouping positions basophils in the upper right quadrant and, unlike gating, allows us to observe interesting properties, such as frequencies and intensities of a third marker color-coded per bin in comparison to all other blood cells. In Figure 1A, the frequency of the CD63+ cells is plotted in the bins, and in Figure 1C, the mean signal intensity (MSI) of the markers HLA-DR, CRTH2 and CD32 per bin is depicted, showing the individual ranges of minimum (blue) and maximum bin-intensity (red) for each sample.

Using PRI, the frequencies of CD63-positive basophils were generated and are shown in Figure 1B for three healthy donors (HD) and six peanut (PN) allergic donors. Basophils from the two IgE non-responders are displayed as triangles. By definition, IgE non-responders are allergic patients who show no or low CD63 activation of the basophils upon an ex vivo anti-IgE stimulation of whole blood. The causes are still unclear. Their incidence is defined inconsistently in the literature and is exemplarily estimated to be 10–20% [16]. The results are fully reproducible, and the color-coded bin visualization of a third marker provides excellent intuitive control over basophil grouping. This is particularly important in donor samples such as patient number 3 (P3, Figure 1C) compared to control 2 (C2, Figure 1A), where basophils and dendritic cell (DC) populations are not clearly separated based on FcεRIα and CD123. It is important to note that some conventional flow cytometric markers/parameters for the BAT, such as CCR3, CD203c and SSC, were either not included in the dataset (CCR3 and CD203c) or are generally not measurable by mass cytometry (SSC). The most useful control markers for grouping basophils were the well-established negative marker HLA-DR and the positive marker CRTH2. In addition, we identified CD32 as a new valuable positive marker. All three markers confirmed a good grouping and separation of basophils in all controls and patients (Figure 1B).

#### 3.1.2. Characterization of CD32 Expression in Basophils

A significant benefit of the bin-based PRI approach is the utilization of various statistical calculations for bins and/or quadrants. By plotting different bin statistics with CD32 as the third marker, we discovered a positive correlation of FcεRIα and CD32 in basophils in terms of both intensity and frequency. In particular, plotting the intensities of only CD32+ cells (MSI+) showed a clear correlation (Figure 2A, top row). Interestingly, we also found a negative correlation between CD32 and CD63 in activated basophils, which is best reflected in the frequencies. The bins with the highest CD32 frequencies have low CD63 frequencies and vice versa (Figure 2A, bottom rows).

When comparing the bin-max intensities of CD32 of all donors, we found a tendency towards higher individual CD32 levels in healthy donors compared to PN-allergic patients. Violin-plotting of the CD32 expression of all basophils of all donors (Figure 2B) confirmed the tendency of lower CD32 expression in PN-allergic donors (P1–P4) compared to healthy donors (C1-C3), unless they are non-responders (P5 and P6). The basophils were defined as all cells located in the upper right quadrant of the bin plots (Figure 1A,C and Figure 2A).

Thus, PRI re-analysis of the BAT dataset was also useful for detecting different intensity levels of a marker in different donor groups and intensity correlations between markers.

### 3.2. Own Flow Cytometry Data Validate Re-Analyzed Mass Cytometry Results

#### 3.2.1. New Marker Combination Confirms Its Potential in Bin-Based BAT Analysis

The robust basophil identification was confirmed with our own flow cytometry measurements and bin-based analysis using the same marker combination identified in re-analyzed mass cytometry data. We collected a total of 1139 individual samples from 50 donors with and without ex vivo stimulation with a range of different allergens. Exemplary patient samples show reliably isolated basophils in the upper left quadrant, applying CD123^negativ/low^ and FcεRIα^high^ on the x- and y-axis in frequency bin plots (Figure 3A). HLA-DR was used as an auxiliary negative basophil identification marker on the z-axis that stains other blood cells right below basophils. Alternatively, CD32, an equally effective positive basophil identification marker, can also be used on the z-axis (Appendix A). These exemplary bin plots demonstrate the effectiveness of the described marker combinations by accurately separating basophils from other blood cells for further detailed bin-based data analysis.

#### 3.2.2. Flow Cytometry Data Confirms Correlation of CD32, FcεRIα and CD63 Expression Patterns on Basophils

Using our PRI approach for the bin-based visualization of flow cytometry data, we confirmed the positively correlated expression pattern of CD32 with FcεRIα (Figure 3B, bottom row). After stimulation with αIgE as a positive control and birch pollen extract as an example allergen, we also observed the negatively correlated expression pattern of CD32/FcεRIα with CD63 on activated basophils (Figure 3B, top row). The reverse expression of CD32 and CD63 can be seen particularly well in the depicted MSI+ bin plots, which only show the intensities of the respective positive cells.

The frequency of positive cells for the z-marker is always shown in the basophil quadrant as a red percentage and shows that CD63 in the example plots increases from 1% (unstimulated) to almost 50% (stimulated), while CD32 shows a slight downward trend, decreasing by a maximum of 15 percentage points (Figure 2B).

#### 3.2.3. Tendency of Higher CD32 Expression Values on Basophils from Non-Allergic Patients Compared to Allergic Patients

Similar to the re-analysis of the mass cytometric data, we observed a tendency for higher CD32 signal intensities on basophils from non-allergic patients compared to allergic patients in unstimulated samples (*n* = 71 in each group). The analysis results of 63 experiments, shown in Figure 3C, reveal that in 40 experiments, CD32 signal intensities of non-allergic patients were higher (dark grey bar), while in 16 experiments, they were lower (black bar) than those of patients with allergies (*p* = 0.0006). In seven experiments, CD32 signal intensities were equal between the two cohorts (light grey bar). These flow cytometric findings of CD32 expression intensity on basophils (Figure 3C) point to the tendency of the data in Figure 2B, derived from a very limited patient cohort.

It is noteworthy that during the establishment of the bin-BAT, we decided to reduce the number of markers by replacing CD123 with the antibody-independent flow cytometry parameter side scatter (SSC), which reflects the granularity and internal complexity of cells. This combination of SSC/FcεRIα with two z-markers, CD32 as an auxiliary marker for basophil identification and CD63 as a basophil activation marker ensured the robust performance of our bin-based analysis and enabled the development of an automated analysis workflow (Appendix A).

### 3.3. Successful Automation of BAT Analyses Using PRI Approach

A comparison of the results of the bin-based automatic workflow (details in Appendix A) confirms the useful application of CD32 or CCR3 as an additional third marker in combination with SSC/FcεRIα for the separation of basophils from other blood cells in a bin-based automatic analysis (Figure 4A). Pearson correlation coefficients R^2^ for CD32 and CCR3 were 0.974 and 0.903, respectively (*p*-value ≤ 0.0001 for CD32 and CCR3), showing a similarly effective identification of activated basophils for both markers. Based on these promising results, a larger auto-BAT validation was conducted with CD32 as an additional standard discriminative marker. The comparison of the manually determined frequency of activated basophils with the corresponding frequency in the auto-BAT of 1139 individual samples resulted in a Pearson correlation coefficient of 0.989 with a *p*-value < 0.001. This validation, presented in Figure 4B, confirms and further supports the good agreement of the automatic results with the manual results. However, there are some outliers (6.15%), about half of which (47.14%) were recognized and flagged by the quality mechanisms used, which is of outstanding value in automated analyses. These results indicate that the automatic workflow using PRI, considering the implemented quality controls, produces robust and consistent results and is a suitable tool for analyzing BATs.

## 4. Discussion

Despite its diagnostic potential, the basophil activation test has not yet made its way to broader clinical implementation. The absence of a standardized and reproducible workflow for sample processing and analysis of complex raw data is one of the key challenges preventing BAT from being accepted as a reliable diagnostic tool for allergies [17,18,19,20,21,22,23,24]. Consequently, several broad-based multi-center studies, including the BAT External Quality Assurance Task Force, have been initiated to evaluate reproducibility in sample processing and data analysis [6,25,26,27]. Here, our contribution to the improvement of the BAT is the application of a different analysis approach to search for new suitable markers and to develop a new automatic analysis workflow.

Our new analysis approach PRI is a unique application to analyze and visualize multi-dimensional cytometry data. It uses a binning approach to group events based on expression intensities for x- and y-markers, creating bin plots with various statistical information and intuitive heat maps for z-markers or additional parameters. Advantages over conventional methods include complete reproducibility using defined thresholds and simultaneous visualization of all cells within one bin plot, avoiding sub-gating and the potential loss of crucial cells or small cell populations.

To improve the BAT, we exploited the potential of our analysis approach, we performed a bin-based re-analysis of a public mass cytometry dataset and obtained three main results. Firstly, we identified an optimal marker combination with FcεRIα^high^ and CD123^+^ to separate and CD63 to monitor activated basophils. Interestingly, Behrends et al. have already described and used FcεRIα as a good marker together with CD203c and CD63 in a recent study [25]. Here, we identified CD32 as a novel auxiliary basophil identification marker using the binning approach of PRI, which consequently led to the concept of an own automation workflow. Secondly, using PRI visualization, we revealed the negative correlating expression pattern of the granule release marker CD63 with FcεRIα/CD32 on activated basophils. These results were also confirmed in our own flow cytometry-based BAT data, suggesting that activated (CD63+) basophils tend to have a lower expression of the inhibitory receptor isoform CD32B and vice versa. CD32B is the predominant isoform expressed on human basophils and, therefore, mainly bound by the FUN-2 clone specific for CD32A/B [28,29,30,31]. Mechanistically, this observation could be seen as in accordance with previous studies describing CD32 as a major suppressor of basophil degranulation as well as of mast cells [32]. Thirdly, we found higher CD32 surface expression values on basophils from non-allergic patients in contrast to patients with allergies in the re-analyzed mass cytometry data set, as well as in a bigger cohort in our own flow cytometry data. These findings contradict the results of a study with 46 patients with Hymenoptera venom allergy and 19 non-allergic patients showing that allergic subjects had a significantly higher CD32 level [33]. Therefore, the relevance of the different basal CD32 expression levels between allergic and non-allergic patients and their possible functional importance should be further investigated in future studies.

An automated BAT analysis generally offers several advantages, such as improved reproducibility, higher efficiency, robust quality control, time and cost savings and standardization even across multiple laboratories. However, the complexity and variability inherent in flow cytometry data make automation difficult, requiring advanced algorithms and thorough validation to ensure accurate and reliable analysis [17,18,19,20]. So far, there are two remarkable automation approaches by Patil et al. [34] and Behrens et al. [25] that both utilize automated gating strategies using Bioconductor tools in the R environment.

We chose an alternative approach to avoid some general difficulties associated with sub-gating methods, which can lead to limited flexibility, data loss and lower reproducibility. Our bin-based auto-BAT workflow instead uses a flexible grouping of cell subsets based on statistical bin data with appropriate markers and preserving all cells [7,8]. Using FcεRIα and SSC as x- and y-planes and CD32 or CD63 as z-markers in multi-parametric binning, we have ensured precise basophil isolation with detailed threshold settings. In addition, robust quality control measures that automatically detect potential problems, such as low event numbers or irregular threshold settings, help to reduce errors to a minimum and improve the accuracy of results. Thus, our automated bin-BAT analysis offers significant advantages for efficient application in both research and clinical settings.

## 5. Conclusions

In this study, we present an automated data analysis workflow for the BAT based on our PRI binning approach, which enables reproducible and reliable basophil identification. The workflow provides a unique visualization of the test results as bin plots, allowing for the intuitive visual evaluation of the automated analysis outcome if desired. Furthermore, quality assurance is enhanced through various quality checkpoints that can inform the user with different warning symbols and messages about potential irregularities in the raw data or the analysis algorithm. Based on this, manual correction of selected samples is possible. In particular, the bin-based information on intensities of z-markers provides valuable information in PRI that goes far beyond sub-gating strategies and helps to distinguish basophils from other blood cells. FcεRIα and CD32 proved to be a very effective pair for basophil identification together with the activation marker CD63, as they show an interesting and reliable pattern.

Altogether, our bin-based automation approach enables a standardized workflow, including a sophisticated system of quality controls, which are an essential prerequisite for exploiting the diagnostic potential of the BAT for public health.

## Figures and Tables

**Figure 1 diagnostics-14-01959-f001:**
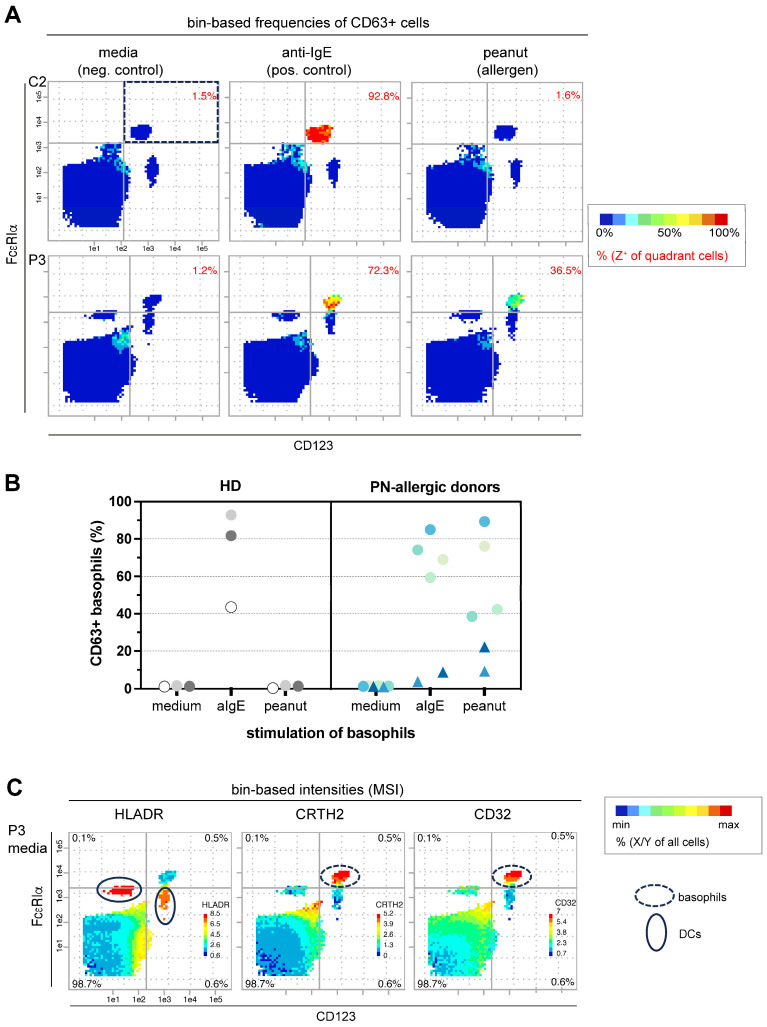
Bin-based re-analysis of basophil activation. BAT mass cytometry cell data [10] were plotted in bins using CD123 and FcεRIα as x- and y-planes to separate basophils (upper right quadrant) from dendritic (DC) and other blood cells. (**A**) As a z-marker, the frequencies of CD63+ cells per bin were plotted with color coding, exemplified by control donor 2 (C2) and the peanut (PN)-allergic patient 3 (P3). The frequencies of CD63+ basophils are shown in red. (**B**) The frequencies for all donors are summarized. Healthy donors (HD) are represented by grey circles and PN-allergic donors are indicated by blue circles. IgE non-responders are shown as triangles. (**C**) As different z-markers, the cell intensities of HLA-DR, CRTH2 and CD32 are color-coded in the bins, and the frequencies of all cells per quadrant are given as black numbers in the corners. The cell intensities of HLA-DR, CRTH2 and CD32 are color-coded in the bins as different z-markers, and the frequencies of all cells per quadrant are indicated as black numbers in the corners.

**Figure 2 diagnostics-14-01959-f002:**
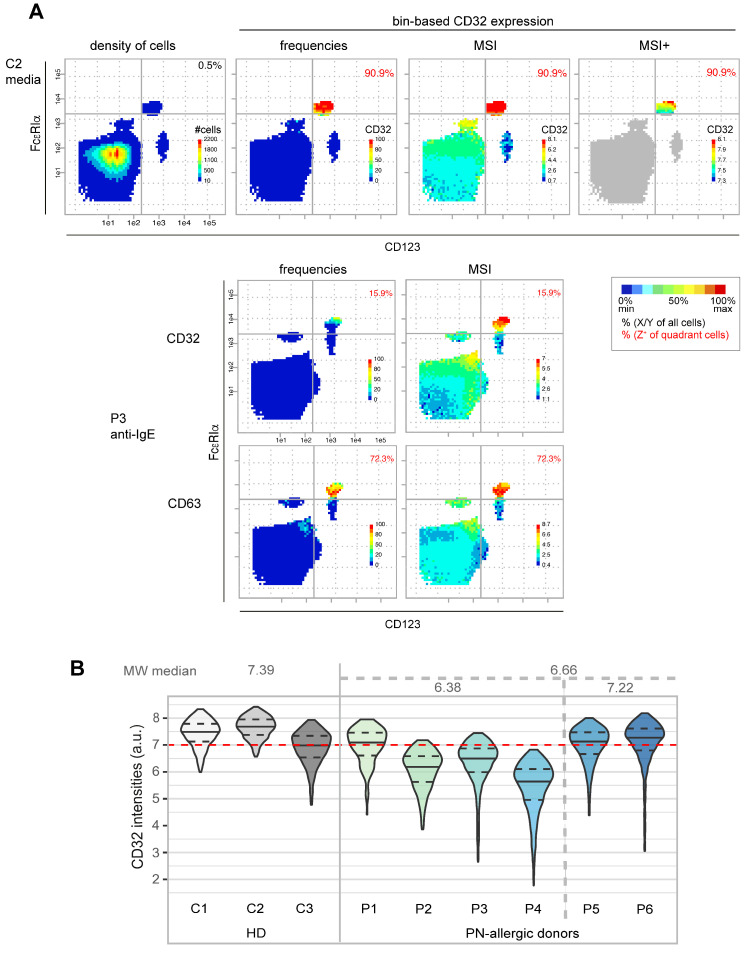
Bin-based analysis of CD32 expression in basophils. (**A**) BAT mass cytometry cell data [10] were plotted with the same x- and y-planes as in Figure 1 with CD32 as the z-parameter. The different color-coded bin statistics are cell density, CD32+ cell frequencies and CD32 mean signal intensities of all cells (MSI) or of only CD32+ cells (MSI+). The frequencies of cells in each quadrant are indicated in black, while the frequencies of the CD32+ basophils are indicated in red. (**B**) The CD32 expression intensity of all basophils from healthy (HD) and peanut (PN)-allergic donors is shown as violins, with 3% outliers removed. Medians are depicted by solid lines, 25% percentile as lower and 75% percentile as upper dashed lines.

**Figure 3 diagnostics-14-01959-f003:**
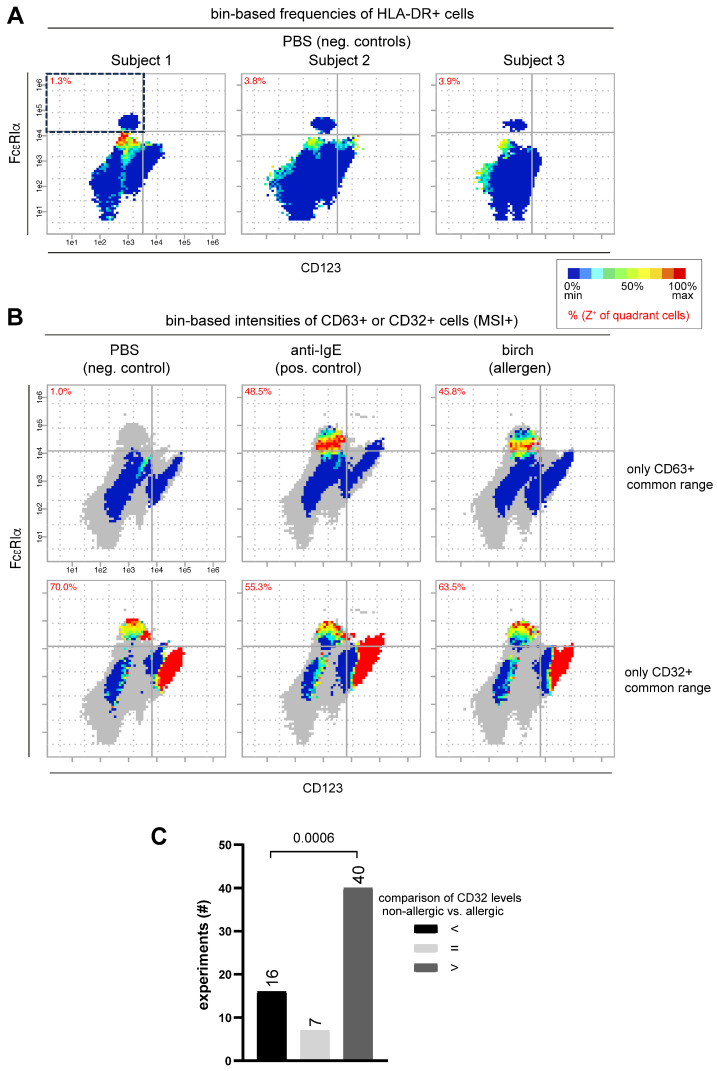
PRI visualization of flow cytometry-based BAT. Three-dimensional bin-based visualizations of basophil separation and basophil activation patterns using specific markers. (**A**) Basophil identification in the upper left quadrant in frequency bin plots using CD123^low^ and FcεRIα^high^ on the x- and y-axis and HLA-DR as an auxiliary marker on the z-axis. Bin colors reflect the frequency in the percent of HLA-DR+ cells within a bin. (**B**) Comparative basophil expression patterns as MSI+ bin plots for CD32/CD63 (negative correlation in stimulated conditions) and CD32/FcεRIα (positive correlation, lower raw). The bin colors reflect the mean signal intensity of cells per bin that are positive for the respective z-marker. (**C**) Statistical comparison of CD32 signal intensities on basophils between non-allergic patients (*n* = 71) and patients with allergies (*n* = 71). Wilcoxon matched-pairs signed rank test: z = −4.23, two-tailed *p*-value = 0.0006, *n* = 63, Pearson correlation coefficient = 0.7261.

**Figure 4 diagnostics-14-01959-f004:**
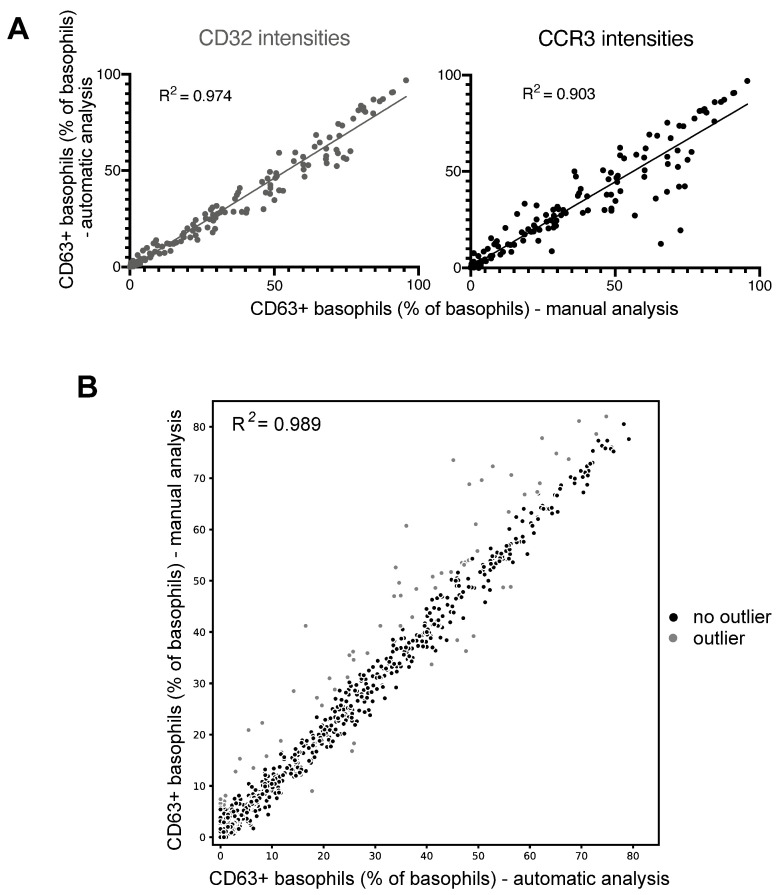
Validation of bin-based automatic BAT analysis workflow. (**A**) Correlation analysis of the frequencies of activated CD63+ basophils (CD63+) using CD32 (left panel) and CCR3 (right panel) as z-markers for bin-based automated BAT analysis compared to bin-based manual BAT analysis. Pearson correlation coefficients were calculated with *p*-values of <0.0001 for CD32 and CCR3. The dataset includes 18 experiments with 195 individual samples measured on two different flow cytometers over a period of 10 months. (**B**) Validation of the bin-based automated BAT analysis with CD32 as an auxiliary z-marker by comparing CD63+ basophil frequencies from the automated workflow with the manual results. Outliers, defined as differences between automated and manual CD63+ intensities greater than Q3 + 1.5 times the interquartile range (IQR), are highlighted in grey. Pearson correlation coefficient was calculated with a *p*-value < 0.001. This dataset includes 50 BATs with 1139 individual samples measured on one flow cytometer over a period of two years and five months.

## Data Availability

The essential functionalities of PRI are available under https://github.com/InesHo/PRI-demonstration, accessed on 1 August 2024.

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
