# Peer review of "Identification of a New and Effective Marker Combination for a Standardized and Automated Bin-Based Basophil Activation Test (BAT) Analysis"

_diagnostics, 2024, doi:10.3390/diagnostics14171959_

Round 1

Reviewer 1 Report

Comments and Suggestions for Authors

Review on manuscript

“Identification of a new and effective marker combination for a standardized and automated bin-based basophil activation test (BAT) analysis”

The research is important and the results are very interesting. The methodological approach is original and may be important for the development of diagnostic approaches.

However, there are a number of remarks

Introduction more than laconic.

Nothing is said about the principle of the Baso-test and the role of investigated markers is not described. (CD63 CD123 CD32)

The authors write that” Recently, we developed a new bin-based approach called “pattern recognition of immune cells” (PRI) for the reproducible analysis and visualization of cytometric data [7-9]. 

-What idea of its approach? This should be added and explained.

In Methods the authors write that “Adult patients with allergies,”

Allergy to what?    Only in fig we can fined PN-allergic donors and in abstract but not in methods. This should be added     And how allergy was confirmed and individuals was selected for study. Patient actual condition and treatment should be added.     

The authors write that “Allergen stimulation was 1 µg/mL birch pollen extract (DST).” 

 -PN-allergic donors what stimulated by birch pollen extract (DST).”?

- 1 µg/mL Is it optimal concentration? and How it was it determined?  

-DST-what is it?  

Extract by what (PBS or alcohol)? And in what (PBS)?

The authors write that “Fresh heparinized whole blood was mixed with RPMI 1640 Medium, GlutaMAX™ 84 (Thermo Fisher Scientific), preheated to 37 °C, containing CD63 antibody and the allergen extract or control”

- How much blood, RPMI and CD63 antibody?

In Fig1.b The authors shown the result of allergen stimulation and Positive control ( αImmunoglobulin E (αIgE) stimulation

As I can understand from figure 1. B  (2 from 6) PN-allergic patient not respond even to positive control it?

This should be explained.

Maine remark

“Identification of a new and effective marker combination for a standardized and automated bin-based basophil activation test (BAT) analysis”

For such an ambitious title, the study group is very small and insufficient.

(peanut allergic patients (n=6) and healthy controls (n=3) 

I’m not sure that (healthy controls (n=3)  its  enough for something like …“pilot study -:CD63 CD123 CD32  combination can increased sensitivity of baso-test in patients with PN allergy“

But still, I am sure that the control group should be larger than the number of co-authors of the article

Author Response

We thank the reviewer for his constructive comments and valuable suggestions, which have significantly contributed to improving our manuscript.

Please find the detailed responses in the attached file as .pdf and the corresponding revisions and corrections highlighted in red and in track changes in the re-submitted files.

Reviewer 2 Report

Comments and Suggestions for Authors

Summary of the Study:

The paper introduces a novel marker combination (FceRIa/CD32) to enhance basophil activation test (BAT) analysis through an automated workflow using a bin-based PRI approach. This method was validated by evaluating 30 markers for optimal basophil identification. Public mass cytometry data, encompassing allergic and non-allergic patients, was reanalyzed and compared with flow cytometry experiments performed by the authors. The findings revealed that CD32 levels can be used to identify allergic patients where they show higher expression in non-allergic patients compared to allergic patients. The study concludes that this new automated analysis method is highly reproducible, can identify small cell populations, improves multi-dimensional cytometry data visualization, and eliminates the need for manual sub-gating.

Comments:

1. The study used 6 samples from peanut-allergic patients and 3 from non-allergic patients from the public mass cytometry database. Is this disparity in sample sizes sufficient for a reliable comparison between allergic and non-allergic groups? Please explain.

2. The figures (Fig. 1B, Fig. 1C, Fig. 2B, Fig. 3) contain numerous abbreviations (e.g., HD, DCs, MSI). To avoid confusion, please define these abbreviations (e.g., healthy donors and dendritic cells) in the text before their first use.

3. In Fig. 1 and Fig. 2, the data comparisons are between Control 2 and Patient 3 samples. In Fig. 2B, comparing C3 and P1 or C1 and P1 might yield different cell frequencies and MSI+ representations, potentially affecting conclusions about CD32+ expression. Were C2 and P3 chosen randomly? Since control and patient samples are mostly not from the same donor, factors such as vaccination or medication could influence CD32 expression in the donor. A more controlled comparison would involve analyzing samples from the same donor before and after allergen exposure. How can the PRI approach address this limitation? Please elaborate.

4. Line 66 references Fig. 3C as data from a cytobank repository, whereas Fig. 3A and Fig. 3B present the experimental results from the authors. This discrepancy is unclear. Please check if this can be included in Fig. 2 rather than Fig.3 .

5. Fig. 3C (showing the number of experiments vs. sample size) may not be necessary in its current format. Could this data be represented differently, such as plotting CD32+ intensity on the Y-axis versus allergic/non-allergic patients on the X-axis?

6. Comparing bin plots in Fig. 2 (Cytobank data) and Fig. 3 (experimental data) shows that basophil populations are not in the same quadrant. Is this variation due to lower CD123 expression in samples exposed to birch pollen compared to peanuts, or is it an artifact of bin-based analysis? Allergens may affect basophil and CD32 expression differently, and individual allergic responses can vary. Does this imply that bin-based analysis might fluctuate with different allergens? How does this impact bin selection and PRI approach? Please comment.

7. The choice of bin can influence how different cells merge with sub-populations, potentially affected by sample purity, staining errors (human errors) and between flow cytometers. How can bin selection be standardized? If variability persists between samples, does this not replicate the issues found with manual gating and sub-gating? Please explain how this variability can be addressed for effective PRI-based analysis.

8. In bin-based plots, based on how bins are defined,  cells near the bin edges can fall into different quadrants, altering analysis results and quadrant percentages. How can this edge effect be mitigated across samples?

Author Response

(The authors gave the same response as above.)

Round 2

Reviewer 1 Report

Comments and Suggestions for Authors

ок